# International Understandings of Health Literacy in Childhood and Adolescence—A Qualitative-Explorative Analysis of Global Expert Interviews

**DOI:** 10.3390/ijerph19031591

**Published:** 2022-01-30

**Authors:** Tessa Schulenkorf, Kristine Sørensen, Orkan Okan

**Affiliations:** 1Interdisciplinary Centre for Health Literacy Research, Faculty of Educational Science, Bielefeld University, Konsequenz 41a, 33615 Bielefeld, Germany; 2Global Health Literacy Academy, Viengevej 100, 8240 Risskov, Denmark; contact@globalhealthliteracyacademy.org; 3Department of Sport and Health Sciences, Technical University Munich, Uptown München-Campus D, Georg-Brauchle-Ring 60/62, 80092 Munich, Germany; orkan.okan@tum.de

**Keywords:** definition of health literacy, childhood and adolescence, global experts, social determinants, policy, contextual level, individual level

## Abstract

(1) Background: With regard to children and adolescents, health literacy should only not be understood as an individual ability, but rather as dependent based on its contextual determinants. The study examines how experts define health literacy in childhood and adolescence and discusses whether they include these factors. (2) Methods: In 48 interviews with experts from 32 countries, specific questions for defining health literacy in childhood and adolescence were analyzed. Data analysis was conducted according to the summary of the qualitative content analysis. Main categories and subcategories were developed exploratively and inductively. (3) Results: No expert had an official definition of health literacy in childhood or adolescence. There were more experts who located health literacy only at the individual level alone than those who located it at both the individual and contextual levels. On the individual level, there was a focus on information processing, knowledge, behavior, and skills. At the contextual level, system responsibility, the ability of others, and relationship between age and development were the main points. (4) Conclusions: To develop an adequate method of dealing with health literacy in the target group, there must be a target group-specific consideration of the dependencies, ages, and developmental stages of that group. While this is considered as consensus in scientific discourse, it has seemingly not yet been adopted in development-related policies internationally.

## 1. Introduction

Nowadays, the meaning of the term “health literacy” seems more complex and heterogeneous than ever. Social scientists argue about what health literacy amounts to. Despite the differences, there is a lot of overlap. Converging developments seem to understand health literacy as a two-sided concept: it pertains to the individual as well as to the structural levels [1]. Underlying this understanding is a shared responsibility between the users and actors in the health system, within which adults are the main protagonists rather than children and adolescents.

With respect to children and adolescents, it may not be sufficient to capture health literacy as a two-sided concept [2]. They are socialized in different environments, contexts, and settings and are particularly exposed to visible (and invisible) dependencies that often do not allow them to make their own health-related decisions. At the same time, they may be ascribed less responsibility than other users of the system. Moreover, the health system’s responsibility toward children and adolescents takes on greater significance since it must consider the specifics of childhood and adolescence [3]. Health literacy interventions must be designed with the needs of the target groups in mind. However, children and adolescents are often addressed as part of the general population. Among other things, this presupposes that the needs of this specific target group are the same as those of adults.

An adequate conceptualization of health literacy in childhood and adolescence can help in designing interventions and policies that are appropriate for the target group. However, an adequate conceptualization can only be achieved by defining and operationalizing the subject matter [4]. Necessary and appropriate health literacy policies can only be constructed when it has clearly been worked out as to what characterizes health literacy in childhood and adolescence. With this in mind, three research questions seem to be central:

1. How is health literacy applied according to international experts?

2. How are children and adolescents taken into account?

3. What unique features regarding children and adolescents are considered?

We will first provide notable theoretical and conceptual considerations of health literacy in childhood and adolescence, which will be followed by a presentation and discussion of the empirical results found in our study. Finally, the findings will be used to draw conclusions for conceptualizing health literacy for children and adolescents.

### 1.1. Health Literacy in Childhood and Adolescence—Contextual Perspective

In a simple approach, health literacy describes a person’s ability to manage his or her own health and navigate the health care system. In this context, health literacy is not static and develops over the course of a person’s life through, e.g., education, and is influenced by personal, situational, and societal factors. It can thus be understood as a dynamic and situation-specific concept at the individual level [5,6]. Most often, the structural and contextual perspective of health literacy is not considered, especially with regard to child and adolescent populations [4,7]. However, this obscures the view of social inequalities in health literacy and the importance of factors that are not under the control of the individual [8]. It is important to consider the extent to which structural, situational, and contextual conditions limit, favor, or change the ability of children and adolescents to act [9]. Bitzer and Sørensen [6] point out that for certain target groups (e.g., children) as well as for those target age groups in certain situations, it is not (only) individual health literacy that determines health and health-related behavior in the long term, but the health literacy of the social environment, e.g., parents, family, circle of friends, or even political leaders (p. 755).

For a long time, children and adolescents occupied little space in the scientific discourse on health literacy [10]. Even today, the amount of available scientific discussion on this topic is rather small [11]. Often, children and adolescents are subsumed under the overall population. It is assumed that the health (information) needs of children and adolescents are the same as those of adults [12]. Children, and especially adolescents, are already confronted with complex health-related information from various sources [11,13]. Especially in this context, it is of increased relevance for children and adolescents to be able to deal with health-related information. Health-related attitudes and behaviors are formed in childhood and adolescence, which can be supported and positively influenced by responding to the information needs of children and adolescents [13].

### 1.2. Health Literacy in Childhood and Adolescence—Definitions and Concepts

In terms of concept developments, Wharf Higgins et al. [14] relate Bronfenbrenner’s [15] socio-ecological model to health literacy in childhood and adolescence. They argue that the health behaviors of children and adolescents are not only individual choices but that they also must include intrapersonal factors at the micro level (e.g., knowledge, skills, and attitudes towards health), interpersonal factors at the meso level (social support and influence from, e.g., family, teachers, health curricula at school), and community factors at the macro level (e.g., health policies, access to the internet).

Based on the initial work of Rothman et al. [16], Okan et al. [17] and Bröder et al. [18] developed a conceptual approach towards health literacy for children and adolescents: the 5D and 6D models, respectively. These models are based on the idea of identifying and focusing on specific characteristics of children and adolescents that are relevant for health literacy considerations and that distinguish children and adolescents from adults.

A review by Bröder et al. [12] addressed the previously developed definition proposals for health literacy in childhood and adolescence. According to the authors, all of the definitions have an action-based focus in common, which primarily encompasses processing health information in various ways. The authors highlight that there is limited consensus on the conceptual underpinnings of the approach and that the specifics of the target group have not yet been elaborated. Comparing the model developed by Bröder et al. [12] with its focus on childhood and adolescence to the more general review by Sørensen [5], the identified definitions overlap and are partly identical. This shows that the specificities of the target group of children and adolescents are not reflected in the definitions.

So far, it has not been examined whether these theoretical constructs have already found their way into political measures to promote health literacy in childhood and adolescence. This is what prompted this study. It became clear that although the discourse recognizes that contextual variables, social and habitual backgrounds, life stages, and individual resources should be included, they are not operationalized and considered in health literacy, and they are particularly not considered when discussing health literacy in childhood and in adolescence. What has been discussed on a theoretical level so far will now be examined at a qualitative empirical level. The question is whether the theoretical considerations are shared by international experts.

## 2. Materials and Methods

Qualitative data were derived from expert interviews conducted as part of the PMO (Project Management Office) policy sub-project of the Health Literacy in Childhood and Adolescence research consortium (HLCA at Bielefeld University). The project focuses on the political actors and professionals who are involved in health policy processes and who can as such provide information on the political implementation of health literacy. The goal of the project is to derive evidence-based knowledge about national and international health literacy policies that can then be incorporated into policy recommendations for political actors. The interview guide comprised a battery of seven questions, including various sub-questions. For the research questions of this manuscript, we only focused on the particular question related to health literacy definitions. The transcripts of the interviews were then reviewed, with an exclusive focus being placed on the second question of the interview: “*What is health literacy in relation to children from your perspective and from the perspective of your country?” Optional addition: “Is health literacy a unique concept or are there other concepts that are similar and currently being addressed in your country?*”. Selected passages were then analyzed by means of qualitative content analysis using MAXQDA software (VERBI, Berlin, Germany).

### 2.1. Interview Study

The experts were contacted via email as part of a literature search and were recruited for the interviews. Expert eligibility was determined through the various personal networks of the project leaders, such as Health Literacy Europe, the Global Working Group on Health Literacy of the IUHPE, the Policy and Advocacy Board of the International Health Literacy Association, and the Asian Health Literacy Association as well as other informal networks and multipliers. Relevant experts were identified through these international networks and based on suggestions made by network and/or association members. Only experts who were working either in governmental organizations or with governmental organizations on developing policies or providing policy work (e.g., as consultants, advisors, expert boards, and/or researchers developing policies for national governments and/or professionals collaborating on policy developments in Non Governmental Organizations) were included. Since health literacy is a relatively new topic in health policy making, not all countries have put health literacy on their policy agendas, and not many policy professionals work on health literacy. Therefore, this was the best way to achieve the main objective, which was to identify those who work or who had worked on health literacy policy. It is important to highlight that the results are not representative of countries but rather describe the state of the conceptual understanding of health literacy for children and adolescents from the perspective of the policy expert and/or that are based on the available country policies.

National and international health policy actors working on health literacy in a ministry, health agency, or NGO were selected. In addition, the experts involved in the development of national health literacy action plans and strategies and in this role working and at the interface of policy, research, and governmental advice were approached. Data were collected through telephone and video interviews conducted by Orkan Okan and Kristine Sørensen. The guide for the expert interviews was available in both German and English.

### 2.2. Sample

The study is based on a convenient sample, and participants were recruited based on their expertise in the field of health literacy policy. A total of 63 interviews were conducted with health policy stakeholders from 45 countries (58 in English and 5 in German). After reviewing all of the interview transcripts, a pre-selection was made, and interviews were excluded that could not be used for the purpose of this study. Specific exclusion criteria were either:

**(a)** No direct answer or statement in response to the question about a definition of health literacy for children and adolescents in one’s own country (e.g., indirect, unintentional mention in a subordinate clause in response to another question)

**or (b)** If a strong interpretation of the statement would have been necessary to extract a health literacy understanding. Accordingly, the inclusion criterion can be formulated as follows: If the question for the definition of health literacy for children and adolescents in the country was answered with a definition or paraphrase of the expert’s understanding.

After excluding interviews that fulfilled criteria (a) or (b), 48 expert interviews from 32 countries remained. In Figure 1, all of the included countries are shown with a light background, and all excluded countries are shown with a dark background. In Figure 1, all of the included countries are in dark-blue, and all of the excluded countries are in bright blue. The country abbreviations agree with ISO 3166 ALPHA-3 and are included after the country name. This is only supplemented by the abbreviations for Northern Ireland, Scotland, and England, which would have otherwise all been counted as the UK.

### 2.3. Data Analysis

The interviews were transcribed and analyzed. Data analysis was carried out according to the summary of the qualitative content analysis process by Mayring [19]. This method is particularly suitable if the development of a category system is to be carried out starting from the material, i.e., inductively, in a systematic reduction process [20]. The suitability of the method is warranted because of this work’s purely explorative intention. The main categories “Individual Level” and “Context Level” were developed inductively. Subsequently, the interviews were divided into segments in order to code the interview passages according to the main categories. From these, various subcategories within the respective main categories were formed during a further inductive step [21].

The individual level was supplemented with various subcategories related to the “handling of information” and “heterogeneous forms of knowledge, skills, and behavior”. Information processing is to be understood from a cognitive psychological perspective. Knowledge, skills, and behavior make up a set of further competencies, as they already frequently serve as collective terms in the literature as well as in empirical studies. The context level was divided into statements related to “interaction with different systems or people”, “provision of information”, and “age- and development-related statements”.

## 3. Results

It should be noted that no expert was able to provide an explicit definition of health literacy for children and adolescents. Rather, most of the experts explicitly expressed that no definition of childhood and adolescent health literacy is available in their country or that no definition is available for the general population. None of the experts indicated that there is a specific definition of health literacy for children and adolescents in their own country; instead, they:Made no statement about it at all;Described their own understanding of health literacy (for children and adolescents);Were not sure if there is an understanding of health literacy for children and adolescent as well as for the whole population;Deflected to descriptions of the concept of health literacy in general.

Thus, no explicit application of health literacy to childhood and adolescence can be recorded for any single nation. It is noticeable that most of the experts made a reference to the fact that they are not working in the field of health literacy regarding children and adolescents (at all or exclusively) and/or that there is no official definition for children and adolescents in the respective country. Several times, this was followed by a reference to a focus on adulthood and/or to the use of only one definition for adulthood or for the whole population of the respective country. Often, either an attempt was then made to apply common adult-age definitions to the target population of children and adolescents, or an explanation is given as to why children and adolescents are also included in a definition for the total population:

“*I don’t know it exactly, but my own definition (…) is pretty similar to adults, but I guess I would say that definition health literacy for children is their ability to find health information, to understand it, to communicate about it. So, evaluate it and then take action to improve their health. So, that for me is the basic definition of health literacy and that would just apply to children as well*”.(*USA I*)

“*I think as for defining health literacy for children and adults, well, or youth (…) they fall within that definition quite well, and so we don’t have a separate definition*”.(*CAN II*)

### 3.1. Individual Level and Contextual Level

Distinguishing health literacy on an individual and a contextual level was a recurring theme. Already during the coding process, it became apparent that the distinction between the broader, contextual perspective and narrower, individual understanding would form one of the core aspects of the data analysis. The individual understanding of health literacy is understood as a focus on information processing or individual knowledge, behavior, or skills. The broader, contextual understanding of health literacy additionally incorporates sociocultural, structural, and societal conditions and resources that enable a person to be more or less competent. It therefore locates competence not exclusively in the individual but also in the environment and surroundings of the individual, as highlighted in the concepts of distributed health literacy [22], social health literacy [23], or organizational health literacy [24]. Figure 2 shows the countries where, according to the experts, health literacy is predominantly considered at the individual level (ARG, BEL, CHE, CHN, EST, FIN, GB, GER, HKG, IND, ISR, KAZ, LBN, MLT, NLD, POL, ROU, SCO, THA, VNM (*n* = 20)) and those experts from countries where both the individual and the contextual levels are mostly considered (AUS, AUT, CAN, ESP, ETH, HUN, HRV, IRL, NIR, NOR, NZL, USA (*n* = 12)). There were no experts who considered the contextual level alone without including health literacy as an individual skill for their country.

#### 3.1.1. Individual Level

Based on the interviews analyzed, the concept of health literacy as an individual and individualized ability can be divided into the handling of information and a number of different knowledge types, behaviors, and skills (see Appendix A, Table A1 and Table A2). Table 1 shows the different emphases related to the individual and contextual levels:

Accordingly, the experts from Belgium, China, England, Germany, Hong Kong, Israel, Kazakhstan, Lebanon, Malta, Romania, and Thailand take an individualized approach to health literacy and focus on information processing and health information use. The experts from Argentina, Estonia, Finland, India, Poland, Scotland, and Vietnam also have an individualized understanding of health literacy but focus on other skills, knowledge types, and behaviors. Experts from Switzerland and the Netherlands put their entire focus on individual health literacy, both on information processing and on knowledge, behavior, and other skills. As expected, experts who consider contextual conditions in addition to the individual level are less likely to focus on individual skills such as information processing or other skills or types of knowledge. The experts from Australia, Austria, Spain, Ethiopia, Croatia, Ireland, and Norway focus on information processing, while only New Zealand’s expert focuses on other skills. The experts from Canada, Hungary, Northern Ireland, and the United States address both areas. From this, we can expect them to reappear in the analysis of the contextual level.

#### 3.1.2. Contextual Level

While the consideration of contextual variables has become a standard requirement in the scientific discourse on health literacy, it has not yet fully entered known definitions and implementation measures. As shown, the experts from 12 nations (AUS, AUT, CAN, ESP, ETH, HUN, HRV, IRL, NIR NOR, NZL, USA) can be grouped together in the category of “health literacy at the contextual level” based on their statements. This category includes statements that do not locate responsibility for healthful action exclusively with the individual but also attributes it to contextual variables such as the health care system. Some statements did not refer to all of the factors considered at the contextual level but rather to certain aspects of it. This included statements that had to do with age- and development-related factors and how they as well as other factors interact. This was the case when interactions with family or parental health literacy were mentioned but, again, this was framed solely as an individual ability (of another person). This was also the case when age- and development-related variables were mentioned in the context of an individual’s developing abilities or limitations.

When the respondents referred to children and adolescents directly, usually the parents, the guardian, and relevant adults from the child’s or adolescent’s environment are being addressed. Accordingly, parental health literacy also seems to play a special role, as it is repeatedly pointed out that children and adolescents are often not understood as independent decision-makers:

“*In adults, it’s about the ability to obtain, understand and use information to make a good decision about your health. So, it’s obviously a lot more complicated in children because they are not necessarily the guardians of their own/they cannot always do what they want to do or, you know, sometimes decisions are made FOR them by other people*”.(*GB I*)

Similarly, it is repeatedly pointed out that a definition for children and adolescents only works when age and developmental stage are taken into account:

“*[T]hat is the definition for the whole population about finding, understanding, appraising, applying health information. And I think that definition might also be relevant for children, depending on their age and cognitive development and health information adapted to children’s age and cognitive development*”.(*NOR I*)

Three superordinate categories could be inductively recognized in relation to the contextual level of health literacy (see Appendix A, Table A3, Table A4 and Table A5):

**Systems in responsibility—provision of and access to information**: When the experts from each country addressed the responsibility of systems, they mostly focused on the way information is provided and accessed. It was made explicit that it must be made easier for children to find, understand, evaluate, and apply health information. Children and adolescents cannot be assumed to understand health-related information made for adults. The health care system should also consider situational factors and the individual circumstances of a person (i.e., children and adolescents). In addition, the way information is communicated, in written text and its visual presentation, are considered to be important.

**Health literacy as the ability of others****:** It is noteworthy that the understanding of health literacy is indeed recognized as being contextual, particularly in relation to surrounding people. However, even with this contextual understanding of health literacy, the person-centered and individualized approach is still maintained since health literacy is then described as the ability of other people. It is frequently pointed out that health literacy is not assigned to children themselves but rather to a family (a group of individuals) or the guardian (or a supporting person). Thus, competence is located in people. Effectively, the responsibility for health-related matters and how to deal with them is left to a child’s social environment. In addition, sometimes children and adolescents are understood as dependent people who cannot make decisions by themselves. Rather, adults make the decisions on their behalf.

**Age- and development-related conditions of health literacy****:** In some interviews, the experts made very specific distinctions in terms of different age groups and developmental stages. This was the case, for example, when they referred to the competencies and knowledge the toddlers, kindergarten kids, schoolchildren, or adolescents should have with regard to their health literacy and the degree of those competencies and knowledge.

## 4. Discussion

The aim of this qualitative study was to explore health policy experts’ and decision makers’ conceptualization of health literacy for children and adolescents. Forty-eight interviews from thirty-two countries were analyzed. The results show that experts define health literacy very differently. Nevertheless, a certain overlap was found in the international operationalizations of health literacy. The commonality among the various health literacy definitions mentioned most often was that health literacy is a concept that is primarily focused on the handling and mastering of health information and how they can be used for decision making and informing healthy behavior. Additionally, health literacy, as a broader set of competencies and skills needed for health promotion, prevention, and health care, also emerged as a recurring theme at all levels throughout the data analysis.

As shown, none of the experts included in this study had an explicit, full-fledged definition of health literacy for children and adolescents as part of an official governmental document or policy. Frequently, the experts made an effort to adapt existing definitions to childhood and adolescence. At other times, they would not tailor them to a specific target group. This may indicate that the scientific discourse on a target group-specific health literacy concept has not yet fully permeated health policy bodies and ministries.

Recalling Bronfenbrenner’s [15] socio-ecological model, it is evident that international health policy actors primarily operate at the micro level. Interpersonal factors at the meso level, for example, support from peers, relatives or schools, are only mentioned in passing in the expert interviews. Macro-level influences, i.e., education, health policy, living conditions, culture, and media, could only be identified in the statements about information provision and presentation.

In the evaluation of the individual competencies, knowledge of health topics was mentioned with above-average frequency. Health literacy seems to be closely linked to knowledge in the understanding of the health policy actors. Knowing how to do something is a prerequisite for actually doing a certain thing, which, in the context of health literacy, means dealing with (health) information, as expressed by almost all of the experts and in line with the scientific discussion [25].

It turns out that “the social environment”, including, e.g., the family, parents, peers, education, and health professionals, was mentioned as a contributing factor. Yet, it remained unclear how much relevance and responsibility can be attributed to each of the factors.

Moreover, adequate communication and the readable and understandable design of information and icons are considered as important. The category “health literacy as an ability of others” bundles the statements that focused on environmental interactions with, e.g., family or peers, and in which these are the people who are supposed to be health literate. This is consistent with the findings of Edwards et al. [22], who used distributed HL to draw attention to the fact that exposure to health information is not an individual’s task but is distributed across social networks (friends, family, work colleagues).

The macro level was mentioned in one of two ways: On the one hand, it was included in statements about the provision of information by the health system. On the other hand, it was mentioned with regard to the responsibility of other (social) systems, for example, formal learning, in teaching and promoting personal health literacy. This also means that only behavior-oriented promotion and prevention is targeted and that no organization-oriented variables are considered. As such, although different settings and systems can be designed to promote health and although health literacy can be promoted and enabled on the organizational level, this has been collapsed into and framed as a mere question of information design on the macro-level.

Generally speaking, measures must be taken so that health literacy can be implemented and developed at the meso and micro levels, within children and adolescents and within their social environment. These measures include the provision of information, education in schools, target group-specific health care services, parental involvement, teacher training and development, professional development, and organizational health literacy [26].

Figure 3 shows the key findings regarding the understanding of child and adolescent health literacy, relevant components, and how often they were mentioned by the policy experts during the interviews. They are structured in accordance with the micro, meso, and macro levels based on Bronfenbrenners socio-ecological model [15] and with the social–ecological framework of adolescent health literacy as introduced by Wharf Higgins and colleagues [14].

The empirical presentation of the results shows a focus on the individual abilities that are relevant to health literacy and the interpersonal factors on the social level of children and adolescents, e.g., parental and professional support and their health literacy. At the same time, the imbalance between personal skills, knowledge, and competencies and the consideration of sociocultural and contextual variables remains, as contextual mentions mostly focused on personal competencies. These findings are similar to those identified by the review conducted by Bröder et al. [12], which focused on the proposed definitions of health literacy in childhood and adolescence that have been developed to date. According to the authors, all of the definitions have an action-related focus in common, which mostly includes the processing of health information. As a central result of their review, the authors state that there is only a limited consensus on the conceptual basis of the concept for children and adolescents and that the specificities of the target group have not yet been considered. Furthermore, individual knowledge- and action-related skills as well as evaluation and critical judgment seem to be the relevant cognitive skills [18]. When comparing the overview of health literacy definitions for the total population [25] to that of health literacy definitions for children and adolescents, the definitions not only overlap, but they are sometimes even partly identical. The specificity of the target population does not figure in the definitions. Furthermore, it is noticeable that the consideration of individual life phase is missing in more than half of the definitions [18].

## 5. Limitations

The expert interviews were conducted with health policy actors and professionals from different countries. Naturally they hardly represent an entire country and, even as experts, they do not have all-encompassing knowledge of health literacy research and its implementation in their respective countries. In addition, the respondents’ statements within the interviews are often to be understood as spontaneous utterances, as the questions were not known to the respondents before the interview. In addition, language can always be a hurdle, especially when conceptual understandings are the focus of the conversation: For respondents whose national language is not English or German, a definition of health literacy in their native language might be more nuanced and may capture the topic better. It is also important to keep in mind that multiple interviews are available from some countries, while only one is available from most. Inevitably, this has the effect of providing a broader view of operationalization for certain countries, limiting and potentially biasing comparability across countries. In addition, the qualitative content analysis evaluation method could only be used in a summarizing and structuring manner. Due to the large number of interviews and the limited space, only one interview question and its pertinent answer could be included in the data analysis. The formation of categories and the presentation of the results pursued the goal of sorting and structuring. Possibly relevant statements made throughout the interviews independent of the second question could not be systematically considered. For this reason and in a next step, it would be important to consider individual interviews in their entirety. Lastly, the experts’ statements about their understanding of health literacy must be placed in the context of a country’s political and health development to ensure a more complete and meaningful classification. However, this was not possible due to the limited scope of this paper and hence awaits future analysis.

## 6. Conclusions

This study shows that while health policy actors define health literacy for children and adolescents very differently, there is some intersection in the ways that it is conceptualized and applied. This is due to the fact that there is no explicit definition for children and adolescents; therefore, prominent definitions for the total population are used. There is overlap in content with regard to the handling of health-related information and for the understanding of health literacy as a set of knowledge, skills, and behaviors. Dealing with health information is operationalized by the experts primarily in terms of understanding and applying, whereas the extended competencies emphasize lexical knowledge about one’s own health. The greatest overlap consists in an understanding of health literacy in children and adolescents as an intrapersonal skill. Yet, considerations rarely extend beyond *individual* skills. When *contextual* factors are included, they take the form of a responsible provision of information through various systems or as the ability of others in the children and adolescents’ environment.

Already in 2016, Malloy-Weir et al. [27] already pointed out that different definitions are problematic for implementation for policy makers, practitioners, and researchers in practice because it is not clear which definition is the most appropriate and what criteria should be used to select and evaluate it. In addition, this may lead to those working with definitions of health literacy in some way using different understandings of health literacy, resulting in ambiguity. Different concepts may be suitable for different contexts, e.g., a country’s individual operationalization needs to be context-specific. The health needs of different countries vary widely, and so do the needs regarding health literacy concepts for children and adolescents.

To come up with an adequate way of dealing with health literacy in the target group, three requirements based on the experts’ contributions could be identified. There must be (1) target group-specific preparation and presentation of health-related information, (2) consideration of their distinct dependencies on adults, and (3) consideration of ages and developmental stages.

While these perspectives are taken into account in scientific discourse, e.g., by the 6D model, they have seemingly not yet been adopted in policy development internationally. There are approaches to increase the recognition of the importance of the living environment (dependency and power relationships with adults, participatory aspects, and development-specific features have at least been recognized), yet this broader understanding beyond personal performance in prominent definitions is not anchored or being disseminated from the scientific field into political spaces. Structural variables are largely only seen at the health systems level. With respect to children and adolescents, at least parents and guardians should also be considered. Although socioeconomic and habitual factors receive some attention and are perceived as a relevant perspective, they fail to be implemented in publicly effective definitions in spite of the fact that the importance of considering contextual variables is scientific consensus. Efforts to conceptualize and apply health literacy tailored to children and adolescents politically are still in their infancy. Despite the availability of some definitions, e.g., by Bröder et al. [18], they have not yet been adopted by actors in the policy field. In the course of this study, mostly definitions referring to the general population were mentioned and extended to childhood and adolescence. More research is needed to explore whether health literacy can be defined for the population as a whole and make specific target group- adjustments in the design and application of interventions or policy programs.

In future work, an integrated analysis should also consider country-specific policies, particularly health policies, and local, target-group differences. In addition, policy developments related to health literacy need to be considered to ensure that the picture of operationalization across countries is complete.

## Figures and Tables

**Figure 1 ijerph-19-01591-f001:**
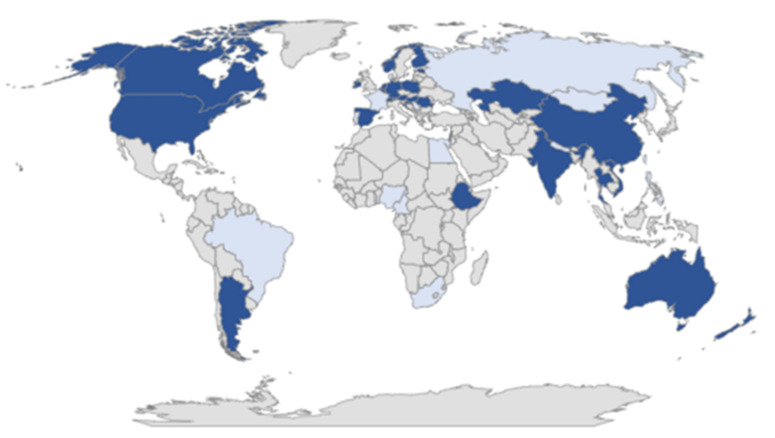
Included (dark blue) and excluded (bright blue) countries for sample analysis: Experts from the following countries were included: Australia (3× (AUS), Austria (AUT), Argentina (ARG), Belgium (BEL), Canada (3×) (CAN), China (CHN), Croatia (HRV), Estonia (EST), Ethiopia (ETH), Finland (FIN), Germany (2×) (GER), Hong Kong (HKG), Hungary (2×) (HUN), India (IND), Ireland (IRL), Israel (ISR), Kazakhstan (KAZ), Lebanon (LBN), Malta (MLT), Netherlands (NLD), New Zealand (NZL), Norway (2×) (NOR), Poland (POL), Romania (ROU), Spain (ESP), Switzerland (3×) (CHE), Thailand (THA), United Kingdom—England (2×) (GB), United Kingdom—Northern Ireland (NIR), United Kingdom—Scotland (SCO), United States (7×) (USA), and Vietnam (VNM). Experts from the following countries were excluded: Brazil, Denmark, Egypt, France, Mongolia, Nepal, Nigeria/Cameroon (both in one interview), Philippines, Portugal, Russia, Serbia, South Africa, and Taiwan.

**Figure 2 ijerph-19-01591-f002:**
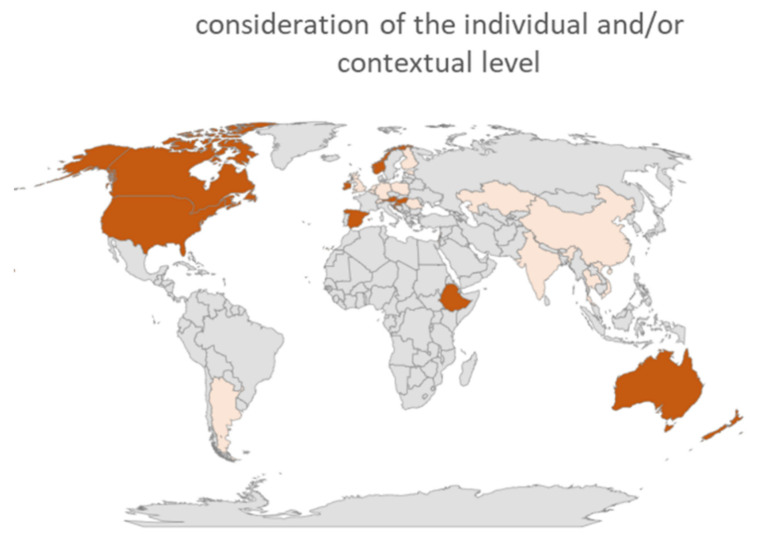
Experts from countries that consider health literacy as an individual concept (light red) or as both an individual and contextual concept (dark red).

**Figure 3 ijerph-19-01591-f003:**
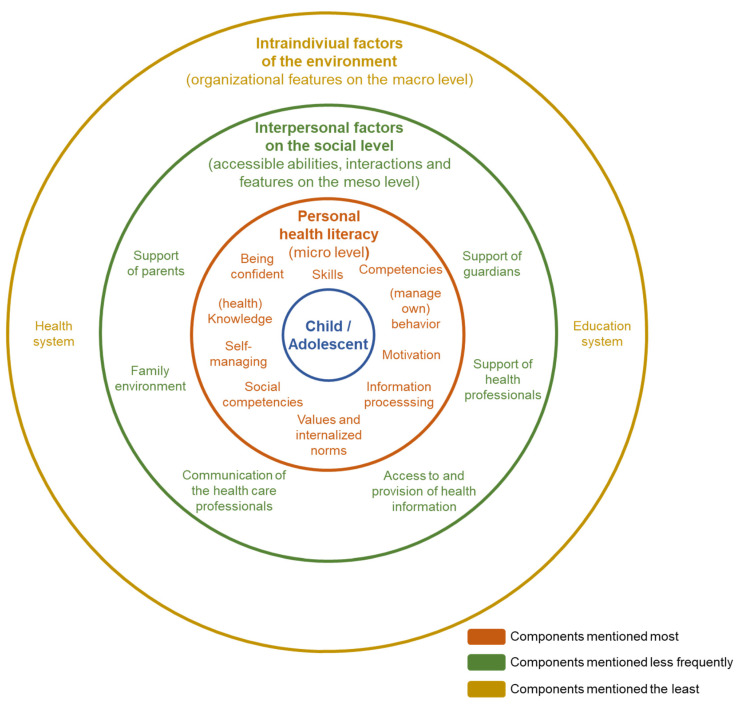
Health literacy defined for children and adolescents—frequency of mentions in the interviews referring to micro, meso, and macro levels based on Bronfenbrenners socio-ecological model [15].

**Table 1 ijerph-19-01591-t001:** Experts from countries that focus information processing and/or knowledge, behavior, and skills.

	Focus on Information Processing	Focus on Knowledge, Behavior, and Skills	Consideration of Both Focuses
**Experts from countries that only consider the individual level**	BEL, CHN, GB, GER, HKG, ISR, KAZ, LBN, MLT, ROU, THA (*n* = 11)	ARG, EST, FIN, IND, POL, SCO, VNM (*n* = 7)	CHE, NLD (*n* = 2)
**Experts from countries that consider the individual AND contextual levels**	AUS, AUT, ESP ETH, HRV, IR, NOR (*n* = 7)	NZL (*n* = 1)	CAN, HUN, NIR, USA (*n* = 4)

## Data Availability

The data presented in this study are available upon request from the corresponding author. The data are not publicly available due to the length of the transcripts and complexity of the qualitative data analysis.

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
