# Peer review of "International Understandings of Health Literacy in Childhood and Adolescence—A Qualitative-Explorative Analysis of Global Expert Interviews"

_ijerph, 2022, doi:10.3390/ijerph19031591_

Round 1

Reviewer 1 Report

The submitted manuscript is well organized and clearly written. This review focuses on the reviewers’ confusion about a conceptual point and then raises a major issue related to sampling and report of findings.

A note about some conceptual concerns: The stated goal of the research is to derive evidence-based knowledge about national and international health literacy policies. However, the questions posed to interview participants from myriad countries focus not on evidence nor on policies but on participants’ definition of terms - health literacy [HL] and adolescent health literacy with particular interest in the unique features of adolescents and children.

The inquiry seems spurred by the WHO’s assertion that “investment in the health 120 and well-being of children and adolescents through improved health literacy is a core 121 public health issue.”  At the onset, the authors discuss the evolving definition of health literacy which considers the interaction between individual skills and structural factors – yielding a shared responsibility between ‘users’ and ‘actors’. Most of the current literature in health literacy does focus on the interactions between adults and professionals/health materials/ and health systems. However, adolescents and children are rarely independently engaged and rarely active decision makers [as the authors note] and they are quite fluid given schooling, skill development, and changes in responsibility. It would be of value to see if and how systems respond to the maturation fluidity of a young person in order to foster health literacy. But a system level response was not the focus on questions posed to participants. Interview participants were asked to define ‘adolescent health literacy’ as though it were a separate concept focused on the characteristics of children/adolescents. 

Methods: A note regarding a major methodological issue concerns sampling.

The manuscript is based on a  convenience sample of ‘political actors’ across and within myriad countries. The authors report on assembling experts – some working in government agencies or NGOs, some responsible for HL reports or programs, others involved in policy. The authors note that ‘eligibility of experts was determined through various personal networks of the project leaders’.  There is no clear articulation of inclusion/exclusion criteria. There is no assurance that the selection process was not biased.

The research is based on interviews with 63 people from 45 countries. Data analysis focused on 48 expert interviews from 32 countries. The authors report that the 15 excluded interviewees either did not respond to the question about definition or responded in such a way that would have demanded a ‘strong interpretation’ – no further explanations were offered.

Participants clearly differed among and within countries in a variety of ways and some countries have had more ‘spokespersons’ than others.  The notion of ‘spokesperson’ is a major problematic issue because data is reported by country as though the perceptions/responses of a selected group of ‘political actors’ within a given country were representative of that country [e.g. lines 354-366]. Of note is that no government documents were submitted or assessed in evidence of policy action or in support of interviewees’ perceptions. 

The sampling method is  profoundly  flawed and the findings are misleading.

The discussion section is quite interesting but the ‘conclusions’ drawn are not supported by evidence based findings.

This exploratory study yielded interesting data. The authors are encouraged to consider crafting a manuscript that accounts for the methodological flaws, addresses issues of definition of terms without attribution to countries rather than to a convenience sample of designated international ‘experts’.  Alternatively, the authors may choose to craft an essay/perspective piece arguing for and perhaps shaping a definition of health literacy that can encompass the needs, abilities, and experiences of young as well as elderly people and offers insights into needed changes in health care and health systems in support of health literacy.

Reviewer 2 Report

This study qualitatively analyses the concept of health literacy in childhood and adolescence. Interviews with health policy experts coming from all over the world were conducted. The manuscript is interesting, and the efforts made are laudable. However, the manuscript is too long with redundant information especially in the introduction and discussion section.

Introduction

Page 2, line 53-55.

The HL measurement methods can be broadly grouped into four categories according to the structure of the tool used: using word recognition items, using reading or numeracy comprehension items (performance-based), using self-reported comprehension items, or using a mixed method (i.e., combination of self-reported and reading or numeracy comprehension items). How you define health literacy, and therefore also how you measure it, is extremely relevant, as you stated. A recent meta-analysis found that when different assessment methods are applied to explore specific HL skills, the prevalence estimates of low health literacy vary significantly (Baccolini V, et al. What is the Prevalence of Low Health Literacy in European Union Member States? A Systematic Review and Meta-analysis. J Gen Intern Med. 2021 Mar;36(3):753-761. doi: 10.1007/s11606-020-06407-8.).

This finding should be mentioned.

The section 1.2 and its subsections 1.2.1 and 1.2.2 are extremely long and dispersive (almost 5 pages of introduction?). Revise these paragraphs to make them more concise.

Methods

Table 1 and figure 1 depict the same information. Keep only one.

Discussion

You are presenting data from 48 interviews, not 62.

This section is too long, and many sentences lack appropriate citations.

Considering the current pandemic, health literacy for adolescents is relevant in relation also to the vaccine uptake and compliance with preventive measures. This should be added. Suggested citations worth including are the following:

  • McCaffery KJ, et al. Health literacy and disparities in COVID-19-related knowledge, attitudes, beliefs and behaviours in Australia. Public Health Res Pract. 2020 Dec 9;30(4):30342012. doi: 10.17061/phrp30342012.
  • Magon A, et al. The effect of health literacy on vaccine hesitancy among Italian anticoagulated population during COVID-19 pandemic: the moderating role of health engagement. Hum Vaccin Immunother. 2021 Oct 13:1-6. doi: 10.1080/21645515.2021.1984123.
  • Turhan Z, et al. The mediating role of health literacy on the relationship between health care system distrust and vaccine hesitancy during COVID-19 pandemic [published online ahead of print, 2021 Jul 22]. Curr Psychol. 2021;1-10. doi:10.1007/s12144-021-02105-8

Subsection 4.2. Where does this come from?

Round 2

Reviewer 1 Report

I appreciate the authors' response and attempt to address issues raised in review. I see that greater specificity was needed.This is a lengthy commentary.

I strongly suggest another round for rewriting and re-submission. I highlight two major areas of concern and offer very specific suggestions. I have taken the effort to do so because I do feel that there is information here that can contribute to the literature.

There are two main points that I highlight here: the first relates to focus and length and the second to the critical issue of appropriately representing findings.

Paper focus and length: The manuscript as submitted veers far from a well organized paper on a study. It combines study information with a far too lengthy lit review, analysis, and editorial [in the opening section, not getting to the point of the study/ in the discussion section reaching conclusions and offering suggestions that are not based on study findings].

I now see on second reading that the authors are essentially combining what should be two papers: an essay/literature review regarding definitions of health literacy [and arguments for a specific definition] and a report/analysis of data derived from interviews conducted with Health Literacy experts from a number of countries. As a result, the lengthy introduction and 'editorial' in the discussion section distract from the analysis - the explorative analysis of global experts' understanding of health literacy in childhood. A  well organized and concise paper focused on the interview data is needed. That is the focus of my comments now. A second paper - an editorial or lit review analysis or perspective focused on definition of terms may well be of interest. [and could cite this paper]. 

I suggest the following: cut lines 65-217, 476-502, 496-547, 586-652

The opening section [lines1-59] set the stage for the effort.

  • I suggest cutting lines 65-217 and preserving this for an interesting literature review and discussion and further suggest moving directly to line 218 to introduce the study.

The Discussion Section - Similarly, the discussion section [which normally focuses on a discussion of the findings and study limitation ] contains material that should be cut as well. Suggestions are offered and conclusions are drawn that do not derive from study findings. This is often considered a 'fatal flaw' in a manuscript submitted for review.

  • I suggest omitting lines 476-502 [and perhaps reserving them for an essay or lit review].
  • In addition, the following discussion paragraphs - line 496-547 are not based on findings from this study but instead are editorial insertions and conclusions [e.g. measures must be taken so that... ] Again, this insightful discussion could well be used in a perspective piece/lit review .... It is not appropriate here. 
  • The discussion section for this study seems to take up again starting on line 553 and ending at line 584 [which should be kept as appropriate for this paper].
  • The paragraphs represented  in Iines 586 -652 seem to introduce new data or new analysis and is confusing in the discussion section. This should be cut.
  • A section on limitations is traditionally included in the discussion section and is warranted here [keep lines 653-677]

Conclusions - A conclusion section generally focuses on implications and calls for further work/inquiry and this section stands.

2. Study findings - I had noted in my first review that study findings were inappropriately  noted by country as though country representatives adequately comprised the study population sample.

The authors did begin to address this issue but did not make all needed corrections. There are several specific places where a misrepresentation still stands [I may not have captured all of them] and findings are reported by country instead of by experts working in a country

  • line 321 - should be no participating expert
  • line 362 - should be participating experts
  • line 367 - should be none of the experts
  • Figure 1: title should be Experts in Countries ...
  • Table 2: should be Experts in ...
  • line 382: should be: accordingly, experts in ...
  • line 385: should be experts in ...
  • ok , I hope I made my point ... see line 391, 393, 400 [experts in ... ], 464 [among experts .. ], 470 [none of the experts]. Please search the text for more.

I do encourage another round of work here and re-submission.

Reviewer 2 Report

--

Author Response

There are no open comments/questions. Therefore, many thanks.